# Working with Deep Generative Models and Tabular Data Imputation

**Ramiro D. Camino** [1]   **Christian A. Hammerschmidt** [2]   **Radu State** [1]

## Abstract

Datasets with missing values are very common in industry applications. Missing data typically have a negative impact on machine learning models. With the rise of generative models in deep learning, recent studies proposed solutions to the problem of imputing missing values based various deep generative models. Previous experiments with Generative Adversarial Networks (GANs) and Variational Autoencoders (VAEs) showed promising results in this domain. Initially, these results focused on imputation in image data, e.g. filling missing patches in images. Recent proposals addressed missing values in tabular data. For these data, the case for deep generative models seems to be less clear. In the process of providing a fair comparison of proposed methods, we uncover several issues when assessing the status quo: the use of under-specified and ambiguous dataset names, the large range of parameters and hyper-parameters to tune for each method, and the use of different metrics and evaluation methods.

## 1. Introduction

Analyzing data is a core component of scientific research across many domains. Over the recent years, awareness for the need of transparent and reproducible work increased. This includes all steps that involve preparing and preprocessing the data. Data with missing values can decrease model quality and even lead to wrong insights (Lall, 2016) by introducing biases. Likewise, dropping samples with missing values can cause larger errors if the amount of remaining data is scarce. One solution is performing data imputation, which consists in replacing missing values with substitutes. However, results often do not hold up if missing

[1]University of Luxembourg, Luxembourg [2]Delft University of Technology, Netherlands. Correspondence to: Ramiro D. Camino <ramiro.camino@uni.lu>, Christian A. Hammerschmidt <c.a.hammerschmidt@tudelft.nl>, Radu State <radu.state@uni.lu>.

*Presented at the first Workshop on the Art of Learning with Missing Values (Artemiss) hosted by the $37^{th}$ International Conference on Machine Learning (ICML).* Copyright 2020 by the author(s).

data is imputed improperly (Lall, 2016).

In this study, we present a short summary for the state-of-the-art of deep generative models applied on tabular data and missing value imputation, and we briefly discuss the problems we faced while working in this field. This falls into three categories: First, the inconsistent use of datasets and ambiguous dataset naming. Second, the metrics used for evaluating the methods. And third, the amount of parameters and hyper-parameters each method can tune.

## 2. Related Work

Work related to this discussion falls into three groups. The first one consists of state-of-the-art imputation algorithms. The second group is composed by generative models based in neural networks, and in particular, networks focusing on generating tabular data and handling issues related to categorical variables, rather than generating one high-dimensional image or text variable. Lastly, the third group is constituted by methodologies using deep generative models for imputation in the domain of tabular data.

Within the field of missing value imputation, traditional methods can be classified into discriminative and generative imputation models. Examples of discriminative models with state-of-the-art performance are MICE (Buuren & Groothuis-Oudshoorn, 2010), MissForest (Stekhoven & Bühlmann, 2012), and Matrix Completion (Mazumder et al., 2010). Expectation Maximization (García-Laencina et al., 2010) is an instance of generative models. Key distinguishing factors of these methods are limitations coming from necessary assumptions about the nature and distribution of the data and the ability to learn from samples with missing data (rather than only learning from complete data samples).

Deep generative models like Variational Autoencoders (VAE) (Kingma & Welling, 2013) and Generative Adversarial Networks (GAN) (Goodfellow et al., 2014) proved to be very powerful in the domain of computer vision (Brock et al., 2018), speech recognition and natural language processing (Lin et al., 2017; Jain et al., 2017). It is expected for the scientific community to extend the application of these models to other areas. The authors of medGAN (Choi et al., 2017) applied GANs to generate synthetic health care patient records represented by numerical and binary

features. The generator in medGAN outputs latent codes that are decoded with a pre-trained autoencoder, and the resulting sample is then judged by the discriminator. The multi-categorical GANs (Camino et al., 2018) extended medGAN and other architectures by splitting the outputs of the networks into parallel layers depending on the size of categorical variables, and used gumbel-softmax activations (Jang et al., 2016; Maddison et al., 2016) to handle discrete distributions. (Mottini et al., 2018) proposed a GAN based architecture to generate synthetic passenger name records, dealing with missing values and a mix of categorical and numerical variables. Tabular GAN (TGAN) (Xu & Veeramachaneni, 2018) presented a method to generate data composed by numerical and categorical variables, where the generator outputs variable values in an ordered sequence using a recurrent neural network architecture.

There are numerous studies related to image completion with deep generative models like (Vincent et al., 2008), that uses denoising autoencoders for image imputing. In the domain of natural language processing, (Bowman et al., 2015) presented a VAE model with a recurrent architecture for sentence generation and imputation. This use case was also translated to the topic of missing value imputation on tabular data. Generative Adversarial Imputation Nets (GAIN) (Yoon et al., 2018) adapted the GAN architecture to this problem. The generator outputs imputed samples from inputs where the missing values were replaced by random noise. The discriminator then tries to predict which values were imputed and which values are original. In Multiple Imputation Denoising Autoencoders (MIDA) (Gondara & Wang, 2018), missing data is initially replaced by the mean or the mode of the corresponding feature and then passed through a denoising autoencoder (implemented with a dropout on the input layer). The reconstruction of the data is considered as the imputed version. To achieve multiple imputation, several models are trained with different random initialization. Importance weighted autoencoders (IWAE) (Burda et al., 2015) presents a generalization of the Evidence Lower BOund (ELBO) defined in VAE that can approximate asymptotically better the posterior distribution. Two studies, Heterogeneous-Incomplete VAE (HI-VAE) (Nazabal et al., 2018) and Missing data IWAE (MIWAE) (Mattei & Frellsen, 2018), extended the work of IWAE to the field of multiple data imputation, by separating variables into missing and observed. Additionally, HI-VAE presents a more extensive collection of losses that depend on the type of each variable.

## 3. Comparison and Concerning Issues

The acquisition of knowledge is an iterative process in the scientific domain. Scientists need to understand, incorporate and challenge the ideas of their pairs continuously. In this context, the ability to reproduce experiments is essential.

However, modern academic times and practices can give birth to publications that leave many details to interpretation. Many seemingly trivial pieces of information might cause discrepancies in our results and wrong conclusions. In our every day work, we found that small details related to the datasets, pre-processing, hyperparameters and the type of reported metric can trigger a cascade of problems.

### 3.1. UCI Repository

In the domain of computer vision there are very well known benchmarks based on datasets like MNIST, ImageNet and CIFAR. However, there is no popular framework to compare studies in the domain of deep learning for tabular data imputation. There is a popular source of datasets called the UCI Repository (Dua & Graff, 2017), but there is no standardized protocol for the usage of this resource. In this paper, we analyze the works on (Yoon et al., 2018; Nazabal et al., 2018; Gondara & Wang, 2018; Mattei & Frellsen, 2018; Hwang et al., 2019). The problem is that the datasets are usually identified with arbitrary short names or aliases which can lead to confusion. For example, one dataset is normally referred as "breast" or "breast-cancer" presents five versions online with different amounts of samples and features. Sometimes the studies refer to the "original" version (Gondara & Wang, 2018; Nazabal et al., 2018), but in other cases is the "diagnosis" version (Yoon et al., 2018; Mattei & Frellsen, 2018; Hwang et al., 2019). Occasionally the authors append tables describing dataset properties like the amount of samples, the amount of features, or the amount of features separated by types (e.g. numerical and categorical). This could solve the issue of dataset identification, but many times the numbers do not match between papers, or with the information provided on the repository. Some datasets like "adult"[1] are originally separated into train and test. Using the entire set or a portion may cause discrepancies in the number of samples reported. The same occurs when the dataset contains missing values and some authors decide to discard them but others do the opposite. In the same way, inconsistencies in the number of features can appear when non-explanatory variables (e.g. ID numbers) are discarded or included in the description tables. The consider that simplest solution would be to identify the datasets by the full URL on the repository. Machine learning software like *scikit-learn* (Pedregosa et al., 2011) provide a collection of datasets to work out-of-the-box, but they are usually considered "toy datasets" because of the dimensions or the complexity of the related task. An ideal approach would be something closer to *torchvision*, the image repository provided by *PyTorch* (Paszke et al., 2017).

---

[1]http://archive.ics.uci.edu/ml/datasets/adult

*Table 1.* Five datasets from the UCI repository including the word "breast" in their names. Note that the number of variables on the repository website may include the target variable or class, and also non-predictive variables like the sample ID.

| FULL NAME | SAMPLES | VARIABLES | | | |
|---|---|---|---|---|---|
| | | PREDICTIVE | NON-PREDICTIVE | TARGET | TOTAL |
| BREAST CANCER | 286 | 9 | 0 | 1 | 10 |
| BREAST CANCER WISCONSIN (ORIGINAL) | 699 | 9 | 1 | 1 | 11 |
| BREAST CANCER WISCONSIN (DIAGNOSTIC) | 569 | 30 | 1 | 1 | 32 |
| BREAST CANCER WISCONSIN (PROGNOSTIC) | 198 | 33 | 1 | 1 | 35 |
| BREAST TISSUE | 106 | 9 | 0 | 1 | 10 |

*Table 2.* Imputation studies using deep generative models that mention the "breast" or "breast cancer" dataset from the UCI repository. The source indicates where details about the dataset (number of samples and variables) were found.

| STUDY | SOURCE | SAMPLES | VARIABLES |
|---|---|---|---|
| GAIN | SUPPLEMENTARY MATERIALS | 569 | 30 |
| HEXAGAN | SUPPLEMENTARY MATERIALS | 569 | 30 |
| MIWAE | CODE EXAMPLE (SCIKIT-LEARN) | 569 | 30 |
| HI-VAE | PUBLICATION | 699 | 10 |
| MIDA | PUBLICATION | 699 | 11 |

## 3.2. Pre-Processing and Metrics

Even if it is not always necessary, scaling numerical features is considered a good practice for training deep learning models. The procedure usually affects the convergence of the training loss, but also changes the magnitude of some metrics. In the case of missing data imputation, it is not clear for some studies if the reported results are measured over scaled or raw features. Additionally, there is a variety of ways in which authors measure the difference between imputed missing values and the ground truth: mean squared error (MSE) (Mattei & Frellsen, 2018), root MSE (RMSE) (Hwang et al., 2019; Yoon et al., 2018), normalized RMSE (NRMSE) (Nazabal et al., 2018), $RMSE_{sum}$ (Gondara & Wang, 2018), etc. Furthermore, the encoding of categorical variables can also affect the experiments but it is rarely mentioned. One could assume that one-hot-encoding is more reasonable because it makes no sense to measure a distance between two label-encoded categorical variables, since the numbers assigned to each category are arbitrary and they posses no valid order. Nevertheless, a one-hot-encoding can increase significantly the amount of feature dimensions, leading to many problems. Besides, deep learning libraries like PyTorch (Paszke et al., 2017) and Tensorflow (Abadi et al., 2016) contain embedding layers for categorical variables that expect label encoded inputs, but categorical outputs are usually handled with softmax layers that involve one-hot-encoding. In any case, note also that one-hot-encoded categorical variables and min-max scaled numerical variables are contained in the range $[0, 1]$, but in other combinations of formats, if the ranges are different, then losses like MSE might assign different weights to each

type of variable during training. If all the variables are in the same range, another possibility could be to aggregate different losses per variable type like in (Camino et al., 2020; Nazabal et al., 2018) but it is harder to implement.

## 3.3. Hyperparameters

Imputation software like *mice*[2], *missMDA*[3], *missForest*[4] or even some modules of *scikit-learn*[5] offer the possibility to run quick experiments by using default hyperparameters. In most of the deep learning related papers, the set of hyperparameters is not indicated, or only partially defined. Furthermore, even if some values are presented, the logic behind the decision is rarely explained. There are settings that need to be defined for most of the deep learning setups: the batch size, the learning rate, the amount and the size of hidden layers, the optimization algorithm (and extra hyperparameters that it may have), decide if dropouts are used, decide if batch normalization is used and decide if any kind of parameter normalization is used (plus related weights). Additionally, for any GAN derived model, the amount of steps each component is trained needs to be defined, and for any model that involves a latent space, the size also needs to be defined. Then regarding more specific models: the parameter clamp for WGAN, the gradient penalty weight for WGAN-GP, the hint probability and the reconstruction loss

---

[2]https://github.com/stefvanbuuren/mice
[3]http://factominer.free.fr/missMDA/
[4]https://github.com/stekhoven/missForest
[5]https://scikit-learn.org/stable/modules/classes.html#module-sklearn.impute

*Table 3.* Common hyperparameters for five different studies. Information marked with (*) was not found in the publication but taken from online code examples, information marked with (?) was not found, and information marked with (-) is not needed for the study. The variable $d$ indicates the dimension of the feature space on each experiment. The list of hyperparameters per model is not comprehensive.

| HYPERPARAMETER | GAIN | HEXAGAN | MIWAE | HI-VAE | MIDA |
|---|---|---|---|---|---|
| BATCH SIZE | 64 | 64* | 64* | 1K | ? |
| EPOCHS | 10K | 3K | 600* | 2K | 500 |
| LEARNING RATE | $1e^{-3}*$ | $2e^{-4}*$ | $1e^{-3}*$ | $1e^{-3}*$ | ADAPTIVE |
| HIDDEN LAYERS | $d, d/2, d$ | $d, d/2, d$ | $128, 128, 128$ | - | $d+7, d+14$ |
| HIDDEN ACTICATION | TANH | RELU | TANH | - | TANH |
| GEN/DISC STEPS | 1/1* | 1/1* | - | - | - |
| LATENT SPACE SIZE | - | - | 10 | 10, 5 | $d+21$ |

weight of GAIN, and the temperature of gumbel-softmax.

It could be argued that some of these choices are not part of the hyperparameter selection but part of the architecture design. For example, the use of *LeakyReLU* in the discriminator (Radford et al., 2015). There are also implementation decisions that can change results and are rarely documented. For example, there is a wide collection "training tricks" for GAN, some of which can be found in (Salimans et al., 2016).

With this large amount of decisions to make, even running a grid search with a small amount of alternative values for each hyperparameter could cost a considerable amount resources. Hence, if the authors of other models do not specify clearly their configurations (or do not publish the entire code for their experiments), and an exhaustive search is not possible, selecting arbitrary values seems very tempting. However, one might assume that researchers put the proper effort for the search of hyperparameters of their own proposals. Additionally, the capacity of different models (the amount of trainable parameters or weights) is not taken into account in many studies. For these reasons, one might suspect that in many articles the competition between models is unfair. The authors of (Lucic et al., 2018) suggest that with enough capacity and training time, many different GAN alternatives can achieve the same results regardless of how complex they look. Also in the metric learning domain (Musgrave et al., 2020) shows that even if articles across the years claim that the field is going steadily forward, some methods from over a decade ago can be competitive nowadays by using them properly and measuring the right thing. Useful comparisons do not only need to list all parameters used for the proposed method, but also tune all baseline and competing methods' parameters.

### 3.4. Simulating Missing Values

In practice, imputation methods are useful when working with datasets containing missing values. However, in order to develop imputation methods, it seems reasonable to deal with datasets where the missing values are actually known and compare them with imputed values. Simulating missing

values can be implemented simply by generating a binary mask that indicates which variables per sample are observed or missing. How this mask is generated can be classified in three ways. The most common approach consists in removing values from datasets to simulate values missing completely at random (MCAR), by removing variables from samples independently of their own values and the values of other variables. Not many studies deal with other types of missingness. Authors in (Yoon et al., 2018) experiment with data that is missing at random (MAR), by removing variables from samples depending on the missingness and the values of other variables. Some studies (Yoon et al., 2018; Gondara & Wang, 2018) experiment with data that is missing not at random (MNAR), by removing variables from samples depending on their own values. Additionally, attention should be put on *when* the mask is generated. If the mask is generated again after each training epoch, the imputation algorithm would be *cheating*, since it would have access to every value of the dataset and overfit to their values instead of actually learn how to impute. Generating a mask before training is recommended to avoid this issue, and also to provide a common scenario to compare models.

## 4. Conclusion

In this paper, we presented a short summary for the state-of-the-art of deep generative models applied on tabular data and missing value imputation, and we briefly discussed the problems we faced while working in this field. We argued that the inconsistent use of datasets and ambiguous dataset naming, the inconsistent metrics used for evaluating the methods and the amount of parameters and hyper-parameters each method can tune are obstacles for reproducible research. Most of these problems could be solved by demanding authors to publish the code for their experiments, and explicitly including all the hyperparameters and proper indications to obtain the datasets. Ideally, competitions or challenges could be organized to standardize metrics and datasets. Another proposal, would be an extension of the UCI repository where pre-processed versions of the datasets are available.

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
