# OpenReview forum: "Working with Deep Generative Models and Tabular Data Imputation"
_ICML.cc/2020/Workshop/Artemiss — ICML Artemiss 2020_

### Official Review · AnonReviewer1 · 2020-06-22
**A wake-up call for writing accurate papers**

**Confidence:** 4
**Rating:** 6

**Review:**

In this paper the authors discuss several issues that arise when comparing deep generative models for missing data imputation from recent research. The authors give examples from recent papers indicating issues regarding dataset identification, inconsistent usage of metrics, hyperparameter selection, etc.

Most of the raised issues can be resolved by releasing source code for papers and making sure that all details necessary for reproducing experiments are given in a paper. In that sense, the findings of the paper do not only apply to papers on topics around missing data but to any machine learning paper. While the authors are certainly correct in their observations, the paper would have been much stronger if they had highlighted the impact of these issues in a few examples. Nevertheless, the authors raise points that we should always have in the back of our heads when writing ML papers.

Minor remark:
Is there a better title for section 3?

---

### Decision · Program_Chairs · 2020-07-02

**Decision:**

Accept

**Comment:**

We are very happy to inform you that your paper has been accepted for the Artemiss workshop. We will contact you soon to inform you about the details concerning the format of your presentation at the workshop, and the camera-ready version deadline. Please take into account the referee's comments to write the camera-ready version.